# Right-Hand Side Expanding Algorithm for Maximal Frequent Itemset Mining

**Yalong Zhang [1,\*], Wei Yu [1], Qiuqin Zhu [1], Xuan Ma [2] and Hisakazu Ogura [3]**

[1] College of Electrical and Information Engineering, Quzhou University, Quzhou 324000, China; 37012@qzc.edu.cn (W.Y.); zhuqq@qzc.edu.cn (Q.Z.)

[2] Faculty of Automation and Information Engineering, Xi'an University of Technology, Xi'an 710048, China; maxuan@xaut.edu.cn

[3] Graduate School of Engineering, University of Fukui, Fukui 910-8507, Japan; ogura@u-fukui.ac.jp

[\*] Correspondence: 37088@qzc.edu.cn

**Abstract:** When it comes to association rule mining, all frequent itemsets are first found, and then the confidence level of association rules is calculated through the support degree of frequent itemsets. As all non-empty subsets in frequent itemsets are still frequent itemsets, all frequent itemsets can be acquired only by finding all maximal frequent itemsets (MFIs), whose supersets are not frequent itemsets. In this study, an algorithm, named *right-hand side expanding* (RHSE), which can accurately find all MFIs, was proposed. First, an Expanding Operation was designed, which, starting from any given frequent itemset, could add items using certain rules and form some supersets of given frequent itemsets. In addition, these supersets were all MFIs. Next, this operator was used to add items by taking all frequent 1-itemsets as the starting point alternately, and all MFIs were found in the end. Due to the special design of the Expanding Operation, each MFI could be found. Moreover, the path found was unique, which avoided the algorithm redundancy in temporal and spatial complexity. This algorithm, which has a high operating rate, is applicable to the big data of high-dimensional mass transactions as it is capable of avoiding the computing redundancy and finding all MFIs. In the end, a detailed experimental report on 10 open standard transaction sets was given in this study, including the big data calculation results of million-class transactions.

**Keywords:** association rule; frequent itemset mining; big data; maximal frequent itemsets

## 1. Introduction

Association rule mining, a research hotspot in recent years, has been widely applied in the era of big data (e.g., cause analysis of traffic accidents [1], association analysis of weather forecasting [2], interest-based real-time news recommendation [3], recommendation of bank marketing plan [4], recommendation of e-commerce matching purchase and shopping basket analysis [5]). In particular, it has been used to realize the data-driven optimization of complex systems in various industries [6].

Association rule mining refers to finding implications such as $A \Rightarrow B$ from the given transaction set, where $A$ and $B$ are itemsets. The former is called an association rule antecedent, while the latter is called an association rule consequent. If the probability for $A$ and $B$ to appear in the transaction set is greater than a certain threshold, it is then called a frequent itemset; otherwise, it is referred to as a non-frequent itemset. $|A|$ or $|B|$ represents the number of items in the itemset. If $|A| = k$, $A$ is called $k$-itemset. Hence, any item can be independently called a frequent 1-itemset or non-frequent 1-itemset. However, the feature of big data association rule mining is that the transaction number of a given transaction set is massive, and the dimension of the transaction set is very high, so it is difficult to mine association rules from it.

$I = \{I_1, I_2, \ldots, I_m\}$ is the set of all items. $D = \{T_1, T_2, T_3, \ldots, T_n\}$ is the big data transaction set with association rules to be mined, where $m$ is the dimension of the transaction set

and $n$ is the number of transactions in transaction set. $T_i \subseteq I$, $T_i \neq \emptyset$, and $i = 1, 2, \ldots, m$. $T_i = \{I_{i1}, I_{i2}, I_{i3}, \ldots \}$ is a transaction in the transaction set. If $A \Rightarrow B$ is an association rule, $A \subset I$, $B \subset I$, and $A \cap B = \emptyset$. The support degree (*support*) and confidence level (*confidence*) are used in association rule mining to measure the weakness of one rule, which are defined as follows:

$$support(A \Rightarrow B) = P(A \cup B) \tag{1}$$

$$confidence(A \Rightarrow B) = \frac{P(A \cup B)}{P(A)} \tag{2}$$

where $P(X)$ is the probability (*support*) for itemset $X$ to appear in given transaction set $D$. Therefore, the corresponding support degree and confidence level of association rule $A \Rightarrow B$ can be calculated only if $P(A \cup B)$ and $P(A)$ are obtained, and whether the two are strongly associated can be determined according to the threshold. For example, assume that $A$ is the itemset $\{1, 2\}$ and $B$ is the itemset $\{2, 3, 4\}$. Then, *support* $(A \Rightarrow B)$ is equal to the probability of the $\{1, 2, 3, 4\}$ appearing in transaction set $D$. *confidence* $(A \Rightarrow B)$ is equal to the ratio of the probability of the $\{1, 2, 3, 4\}$ to probability of the $\{1, 2\}$ in transaction set $D$. Hence, the association rule mining problem can be transformed into a mining problem of frequent itemsets.

As frequent itemsets have a very important property (i.e., the non-empty subsets of all frequent itemsets are also frequent itemsets, and the supersets of all non-frequent itemsets are also non-frequent itemsets), if an itemset is a frequent itemset but any of its superset is a non-frequent itemset, it contains most frequent itemsets with the maximum capacity. In this way, the algorithm is able to find all frequent itemsets with the least optimization objectives. The itemsets of this type are called maximal frequent itemsets (MFIs). Thus, the frequent itemset mining (FIM) problem can also be transformed into the problem of maximal frequent itemsets mining (MFIM).

For the purpose of association rule mining, MFIM has some advantages over FIM. Firstly, it makes the optimization problem have fewer targets to be searched. For a set of transactions with a given support threshold, the set of all MFIs is a subset of the set of all frequent itemsets. The number of elements in a subset is always less than the number of elements in the set, so there were fewer targets for optimization. This helps to speed up the running time of the algorithm. Secondly, not all frequent itemsets are useful in calculating association rules. The algorithm avoided reinventing the wheel. The disadvantage of MFIM is the theoretical addition of frequent itemsets generated by MFI. However, in fact, the long pattern MFIs found were not very long in practice, and it is very easy to generate the required frequent itemsets. This disadvantage is almost negligible compared with the benefits brought to the algorithm.

In addition, both MFIM and FIM can not only be used to calculate association rules such as (1) and (2), they also have many other applications. For example, they can be used for outlier detection [7], which is a kind of data mining technique to detect rare events, deviant objects, and exceptions from data, which has been drawing increasing attention in recent years. MFIM and FIM can also be used for web clustering [8]. A vast number of documents in the Web have duplicates, which is a challenge for developing efficient methods that would compute clusters of similar documents. Web clustering can be conducted through FIM. MFIM can also be used for partitional clustering. Dinh et al. [9] took advantage of non-random initialization from the view of MFIM to improve clustering quality. Beyond this, MFIM may have more applications.

However, most of the present studies regarding the association rule mining aim at frequent itemset mining, while MFIs have been scarcely taken as the mining object. The current algorithms specific to frequent itemset mining are largely divided into two major types: exact algorithms and heuristic algorithms. The most classical exact algorithms are the Apriori algorithm [10] and FP-Growth algorithm [11], as well as many improved algorithms derived from the two algorithms [12–26].

Faced with high-dimensional mass big data, the exact algorithm itself is almost of no practicability due to the temporal complexity and explosion of storage space. However,

some calculation platforms that can realize the temporal and spatial decomposition of data mining tasks have emerged in order to process big data, so the exact algorithm becomes feasible [12,27–29]. The advantage of these calculation platforms lies in the fact that a big data analysis becomes feasible due to computer clusters, among which Spark reaches the highest rate at present.

Most of the heuristic algorithms have integrated evolutionary computation [30–32], particle swarm optimization (PSO) [33–35], and other artificial intelligence algorithms with exact algorithms [36]. Bagui et al. [31] applied a genetic algorithm (GA) to data flow to mine frequent itemsets, and the novelty of this work is in the use of frequent itemsets to determine the concept drift. As the object was partial data selected from the data flow using a sliding window, the dimension (number of items) and number of transactions of the processed dataset were small. Sizov et al. [31] also designed a GA to acquire the frequent itemsets and large bite sets of the binary transaction set, and gave the application cases of 23-column 5712-row transaction sets, whose volumes were small. Ykhlef et al. [32] used a quantum evolutionary computation to mine frequent itemsets from the nursery transaction set, comprising 12,960 transactions and a dimension of 32.

Zhang et al. [33] designed a binary PSO algorithm to mine frequent itemsets. This algorithm could realize dynamic pruning during the population initialization and evolution process to relieve the time pressure of memory and CPU, and it was applied to four different transaction sets. The number of transactions was small in all four transaction sets, among which three transaction sets contained 1000 transactions, and one contained 500 transactions. Chiu et al. [34] used the PSO algorithm to mine frequent itemsets in a transaction set called FoodMart2000, which contained 12,100 transactions with a dimension of 34. Kabir et al. [35] enhanced the random search performance of the PSO algorithm and mined frequent itemsets in a transaction set with 1000 transactions and a dimension of 5.

Paladhi et al. [36] designed an artificial cell division algorithm, which was very successful in solving multipath search tasks involving search space and achieved superior effect when applied to small-scale transaction sets compared with the Apriori algorithm.

Although the abovementioned algorithms have not been applied to high-dimensional big data transaction sets, the heuristic algorithm is theoretically feasible in the face of high-dimensional mass big data. Nevertheless, the solution found is an optimized solution but not an exact solution. The exact solution can be found using the exact algorithm, which, however, decomposes a task based on special computing platforms when applied to big data (e.g., Spark). Moreover, exact algorithms are based on cluster computing, which requires multiple computers. Therefore, both algorithms have their respective advantages and disadvantages when solving the big data association rule mining problem.

For large-scale transactions set, parallel data mining is very promising, so a few algorithms for parallel mining association rules were proposed. The most well-known one of them is the Count Distribution (CD) algorithm [37], which is a parallel version of the Apriori algorithm. In CD algorithm, databases are initially partitioned and distributed across multiple processing nodes.

The FPMAX algorithm [38] based on FP-Tree was one of the most efficient and stable mining algorithms for maximum frequent itemsets. However, for mining in dense transaction sets, FPMAX would generate many redundant recursive procedures, resulting in an additional conditional FP-tree construction overhead. Additionally, when the support was low, FPMAX would degrade the performance of superset detection due to the large global MFI-tree used for superset detection.

The parallel max-miner (PMM) algorithm [39] and the distributed max-miner (DMM) algorithm [40], proposed by Soon M. Chung and Congnan Luo, are two excellent MFI mining algorithms for high-dimensional big data. Compared to most of existing mining algorithms, PMM looked ahead at each pass and prunes more candidate itemsets by checking the frequencies of their supersets. DMM has a local mining phase and global mining phase. During the local mining phase, each node mines the local database to discover the local maximal frequent itemsets, and then they formed a set of maximal

candidate itemsets for the top–down search in the subsequent global mining phase. A new prefix tree data structure was developed in DMM to facilitate the storage and counting of the global candidate itemsets of different sizes. This global mining phase using the prefix tree can work with any local mining algorithm. Both PMM and DMM were implemented on a cluster of workstations, and they required very low communication and synchronization overhead in distributed computing systems.

From the above analysis, we propose an exact algorithm. This algorithm does not used any computing platform, nor does it required a cluster of computers, but it could find the exact solution—all MFIs of big data transaction set within an acceptable time range. In the other word, the motivation of the algorithm proposed in this paper is to realize the MFI mining in an acceptable time range without using cluster computers for the big data with high-dimensional attributes and massive transaction numbers.

The algorithm presented in this paper will contribute to the MFIM problem in the following ways. ① The reduced transaction set is generated and used without changing the mining results. In this way, the effective part of the transaction set can be loaded into the RAM to be accessed, which not only accelerates the algorithm speed, but also provides a basis for single-machine computing of high-dimensional big data with a massive transaction number. ② The Expanding Operation adopts the strategy of one-way search so that every MFI can be found and the path to be found is unique. In this way, the computational redundancy of the algorithm is avoided as much as possible.

## 2. Right-Hand Side Expanding Algorithm

In this study, the proposed algorithm was divided into three parts: ① First, the transaction set with mass data was preprocessed. A transverse and longitudinal data reduction was then performed according to the given support threshold. Thus, a simplified transaction set was obtained. The use of a simplified transaction set would not change the algorithm's accuracy. Instead, it could significantly accelerate the computation. ② Second, an any frequent itemset-oriented operator, which could extend and add items from any given frequent itemset so as to find the MFIs, was designed. ③ The itemsets of starting points were reasonably organized and given for the Expanding Operation so that it could find all MFIs.

Above all, the notations, and functions to be used often in the text firstly are listed in Table 1. They will be explained at first use in the text. However, the reader may find it more convenient to look up definitions in the table.

**Table 1.** Notations and functions.

| Notation | Meaning |
| --- | --- |
| $D$ | $D = \{T_1, T_2, T_3, \ldots, T_n\}$ is the given transaction set |
| $I$ | $I = \{I_1, I_2, \ldots, I_m\}$ is the set of all items |
| $m$ | The dimension of the transaction set |
| $n$ | The number of transactions in transaction set |
| *Support* ( ) | Return the support for a given itemset. Additionally, the probability of occurrence for the given itemset |
| *Confidence* ( ) | Return the credibility for given association rule |
| $T_i$ | $T_i = \{I_{i1}, I_{i2}, I_{i3}, \ldots, I_{ik}\}$ represents the $i$th transactions |
| $I_{ij}$ | The $j$th item of $i$th transactions in transaction set |
| *Support_T* | Support threshold |
| $ND$ | The new transactions set returned by the Algorithm 1 |
| $NTi$ | The $i$th transactions in $ND$ |
| $m^-$ | The dimension of the reduced transaction set |
| $n^-$ | The number of transactions in the reduced transaction |
| $k$ | Has been used as the number of items of itemsetIt's also used as a general constant |
| $k^-, k^+$ | Two different natural numbers comparing with $k$, and $k^- < k < k^+$ |
| $P$ ( ) | Returns the probability of a given item set in transaction set |
| $P$ | The base point at which the Expanding Operation adds item |

*2.1. Transaction Set Preprocessing*

The preprocessing of transaction set was conducted to obtain a reduced transaction set, and the original transaction set should be scanned twice. First, the appearance probability (support) of each 1-itemset was acquired. Next, all items with a probability smaller than the threshold were excluded. Thus, a new and reduced transaction set was obtained.

$D = \{T_1, T_2, T_3, \ldots, T_n\}$ is the original big data transaction set, and $n$ is the number of transactions in the transaction set, where $T_i = \{I_{i1}, I_{i2}, I_{i3}, \ldots\}$, $(i = 1, 2, \ldots, n)$ represents the $i$th transactions of the transaction set, $I_{ij}$ represents the $j$th item of $i$th transactions. Each transaction is a set of items is also a subset of set of all items $I = \{I_1, I_2, \ldots, I_m\}$.

If $I_{ij}$ is the *ID* of one item and all of *ID* are continuously numbered together from 1, then,

$$m = Max_{i=1}^n Max\{I_{i1}, I_{i2}, \ldots\} \tag{3}$$

where $m$ is the maximum *ID* of an item in original big data transaction set. By this way, the $m$ is also the dimension of the transaction set because the *ID* is from 1, namely, the value of $m$ directly decides the dimension of the solution space in itemset mining. For example, we assume that the itemset $\{I_{11}, I_{12}, \ldots\}$ is $\{1, 2\}$, the itemset $\{I_{21}, I_{22}, \ldots\}$ is $\{2, 3, 4\}$, and the number of transaction $n$ is 2. Then, $m = Max\{2, 4\} = 4$, the dimension of the transaction set is 4. Under a given transaction set $D$ with the known dimension $m$ and support threshold (*Support_T*), the pseudocode of transaction set preprocessing is expressed as seen in Algorithm 1.

Algorithm 1 contains two two-level nested "for" loops. The first two-level nested loop records the number of occurrences of each item, the inner loop traversed the items of each transaction, and the outer loop traversed the entire transaction set. At the end of the first two-level nested loop, the $m$-dimensional vector $(S_1, S_2, \ldots, S_m)$ recorded the number of occurrences of each item in the transaction set. The second two-level nested loop was used to rebuild a new reduced transaction set. The inner loop was used to rebuild a new transaction that does not contain infrequent items. The outer loop added each new transaction to the newly created reduced transaction set.

The algorithm returned to a new transaction set *ND*, which consisted of new transactions $NT_i$. When a new transaction was generated, it was first made empty, and only the items with a high support degree could join in the new transaction. It was possible that all items in some transactions in the original transaction set $D$ were not frequent items, so they would not be added in the new transaction, and then the new transaction was an empty set before and after the third for loop. As a result, it would not be added into *ND*, the number of transactions in *ND* would be reduced, and the number of transactions in the pseudocode was turned from $n$ into $n^-$, and $n^- < n$. The number of transactions in the new transaction set would be reduced to different degrees due to the change of the data sparsity in the original transaction set $D$.

When a new transaction $NT_i$ was established, some items would be given up, and the quantity of all different items appearing in the new transaction set would be reduced, and so would the dimension of the reduced transaction set. The decreased amplitude of dimension varied with the given support threshold after the reduction. The dimension of the original transaction set is expressed as $m$, and that of the reduced transaction set as $m^-$, and $m^- < m$. The optimization space of the algorithm was greatly reduced due to the dimension reduction, thus elevating the operating rate to a great extent.

---

**Algorithm 1:** Transaction Set Reduction

---

**Input:** $D = (T_1, T_2, \ldots, T_n)$, $T_i = (I_{i1}, I_{i2}, \ldots)$
**Output:** $ND = (NT_1, NT_2, \ldots, NT_{n^-})$
$(S_1, S_2, \ldots, S_m) = (0, 0, \ldots, 0)$;
$n^- = 0$;
for $i = 1$: $n$ do
    for $j = 1$: $|T_i|$ do
        $S_{(Iij)} = S_{(Iij)} + 1$;
    end
end
$ND = \varnothing$;
for $i = 1$: $n$ do
    $NT_i = \varnothing$;
    for $j = 1$: $|T_i|$ do
        if $S_{(Iij)}/n > Support\_T$ then
            $NT_i = NT_i \cup \{I_{ij}\}$;
        end
    end
    if $NT_i \neq \varnothing$ then
        $ND = ND \cup \{NT_i\}$;
        $n^- = n^- + 1$;
    end
end
Return $ND$;

---

### 2.2. Expanding Operation

The expanding operator adds items to frequent itemsets according to certain rules and finds the supersets of some frequent itemsets, all of which are MFIs. The expanding operator has two features: First, the given initial itemsets must be frequent because all supersets of non-frequent itemsets cannot be MFIs. Second, what the Expanding Operation finds are not all MFI supersets corresponding to the given frequent itemsets but partial MFIs. As any MFI can be obtained by expanding different frequent itemsets, the Expanding Operation only finds the partial MFIs corresponding to the given frequent itemsets, and the remaining part is obtained through the other frequent itemsets using the Expanding Operation. This ensures that each FMI is found only once.

The initial frequent itemset $T = \{I_1, I_2, I_3, \ldots\}$ was given, an integer $p$ ($0 < p < m^-$, where $m^-$ is the dimension of the reduced transaction set, namely, the total number of different items) as the reference position for the added items, and then the pseudocode of Expanding Operation was expressed as in Algorithm 2, where $ND = \{NT_1, NT_2, NT_3, \ldots, NT_{n^-}\}$ is the reduced transaction set, $NT_i = \{NI_{i1}, NI_{i2}, NI_{i3}, \ldots\}$ and $i = 1, 2, \ldots, n^-$ are the transactions in the transaction set, and $n^-$ is the number of transactions in the reduced transaction set ($n^- < n$).

The operation contained four "for" loops (nested loops excluded). In the first "for" loop, a frequent itemset $T$ was expressed by a decision variable $(x_1, x_2, \ldots, x_{m^-})$. $x_i = 0$ means that the item $i$ was not a member of the itemset. If $x_i = 1$, the item $i$ is a member of the itemset. The second "for" loop acquired all single items that could be added into the given frequent itemset, and after the addition, the itemset was still a frequent itemset. In the third "for" loop, the number of single items was calculated. If it was equal to the number of items in the frequent itemset, the given frequent itemset was namely an MFI, and then this itemset was returned. Otherwise, the fourth "for" loop should be executed: An addable single item was added rightward from point $P$ by turns. The values of the given initial frequent itemset $T$ and base point $P$ were reset once a single item was added, followed by the recursive invocation of Expanding Operation itself. After all recursive nested structures were returned, all MFIs corresponding to the original given frequent itemset were found. As the operator only added the items at the right-hand side of point $P$ (items with *ID* value greater than the $p$ value) each time, the algorithm was named right-hand side expanding (RHSE) algorithm.

---

**Algorithm 2**: Expanding Operation (*T*, *P*)

---

**Input:** $T = (I_1, I_2, \dots)$, $P$
**Output:** A group of maximal frequent itemsets
$(x_1, x_2, \dots, x_{m^-}) = (0, 0, \dots, 0)$;
$(e_1, e_2, \dots, e_{m^-}) = (0, 0, \dots, 0)$;
for $i = 1$: $|T|$ do
  $x_{(Ii)} = 1$;
end
for $i = 1$: $n^-$ do
  if $T \subseteq NT_i$ then
    for $j = 1$: $|NT_i|$ do
      $e_{(NIij)} = e_{(NIij)} + 1$;
    end
  end
end
$e\_item\_number = 0$;
for $i = 1$: $m^-$ do
  if $e_{(i)}/(n^-) > Support\_T$ then
    $e_i = 1$;
    $e\_item\_number = e\_item\_number + 1$;
  else
    $e_i = 0$;
  end
end
if $e\_item\_number = |T|$ then
  Adds $T$ into group of maximal frequent itemset;
else
  for $i = P + 1$: $m^-$ do
    if $(x_i = 0)\&(e_i \neq 0)$ then
      Expanding Operation $(T \cup \{i\}.$ $P = i)$;
    end
  end
end

---

## 2.3. Overall Framework of Algorithm Running

With this Expanding Operation, it is only necessary to reasonably organize and alternately give the initial frequent itemset of the Expanding Operation, which is then invocated. Thereafter, all MFIs of the given support threshold can be found. Under the overall framework of algorithm running, a reduced transaction set was first obtained after the preprocessing. During the preprocessing, all frequent 1-itemsets were acquired, which were set as the initial itemset alternately. Items were added by invocating the Expanding Operation, and then a group of MFIs corresponding to each frequent 1-itemset could be found. When placed together, these groups were the set of MFIs discovered by the algorithm. These MFIs were neither repeated nor omitted, being the exact solution but not the optimized solution to the problem.

The overall framework of algorithm running was expressed by a block diagram in Figure 1, where $I_1, I_2, \dots, I_{m^-}$ are the $m^-$ frequent 1-itemsets acquired in the preprocessing of the transaction set, which aimed to obtain a reduced transaction set, and $m^-$ is the dimension of the reduced transaction set. As the dimension of the original transaction set is $m$, the dimension of the reduced transaction set is expressed by $m^-$, and $m^- < m$. The $m^-$ frequent 1-itemsets were set as the initial itemset of Expanding Operation, namely, $I_i$, and the base point $P = i$, $(i = 1.2, \dots, m^-)$ was given in turn. The Expanding Operation was then invocated. Subsequently, several MFIs were acquired and called the group of MFIs, and these groups of MFIs finally formed a larger pool of MFIs. This larger pool of MFIs was an exact solution to the problem, and it was a set of MFIs.

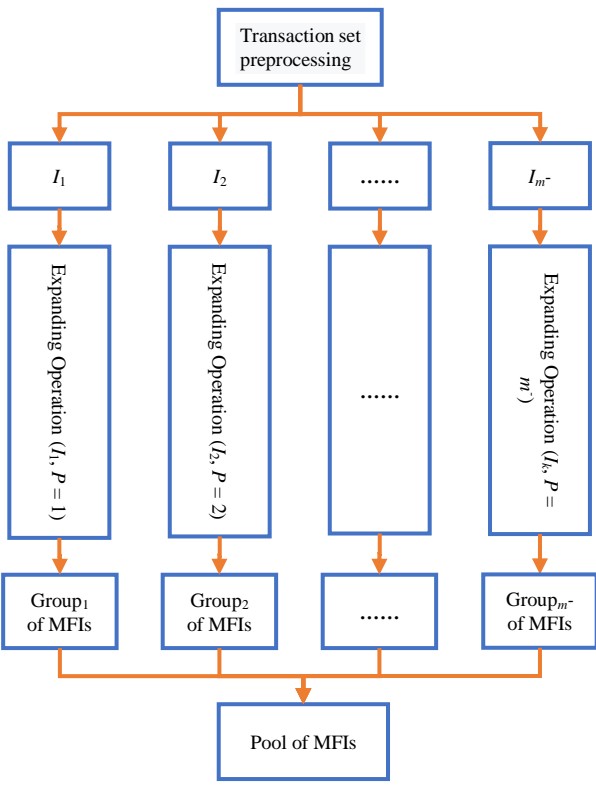

**Figure 1.** Overall framework of algorithm running.

*2.4. Complexity Analysis*

The time complexity of the proposed algorithm RHSE was analyzed with respect to Algorithms 1 and 2. Algorithm 1 was used to generate the reduced transaction set, which requires scanning the transaction set twice. $n$ is the number of transactions in transaction set, and let $L_i$ be the average number of items in the transaction (itemsets). Thus, the time complexity of this task is $O(n \times L_i)$. Algorithm 2 is the *Expanding Operation* with an initial itemset and position point. Each *Expanding Operation* contained $(m^- - (P + 1))$ branch recursions, where $m^-$ is the dimension of the reduced transaction set $(m^- < m)$, and $P$ is the base point at which the expanding operation adds item. Thus, the time complexity of Algorithm 2, *Expanding Operation*, is $O((m^- - (P + 1))!)$. In comparison, the solution space of the problem is $2^m$ and the time complexity of the enumeration method is $O(2^m)$. We will learn in Section 4.3 that the time complexity of Algorithm 2 is much lower than that of enumeration.

## 3. Proof of Algorithm Accuracy

The solution to the optimization problem is divided into two types: an exact solution and optimized solution, both of which are feasible solutions. The corresponding algorithms can also be classified according to the solution type. For instance, in the introduction part, the algorithm acquiring the exact solution was called the exact algorithm, and the algorithm acquiring the optimized solution was called the heuristic algorithm. The exact solution refers to the optimal solution to the problem, and this solution or set of solutions is unique. Here, the solutions may be a set of feasible solutions with equivalent excellence, and the uniqueness means that this set is unique. The optimized solution may not certainly be the optimal solution, but it may also be the optimal, and its excellence is pursued as much as possible. If the solution is a set, this set may have a missing element.

The exact solution to MFI mining is the set of all MFIs found by the algorithm. The itemsets in this set should not be estimates. Instead, this set is the well-determined set consisting of itemsets. The proof of algorithm accuracy aims to prove that under the given

transaction set and support threshold, all MFIs will be found (lack of any itemset is not allowed), and each MFI is found only once (repetition is not allowed). Given this, two problems remain to be proven: ① any MFI can be found, which is called integrity; ② any MFI can be found only once. The MFIs found from the pool of MFIs are not the same; this is referred to as uniqueness.

### 3.1. Integrity Proof

Assuming that $\{I_1, I_2, \ldots, I_k\}$ is a random MFI, the corresponding decision variable is $X = (x_1, x_2, \ldots, x_{m-})$, and $x_i \in \{0,1\}$. $x_i = 0$ means that the item $i$ is not a member of itemset, and $x_i = 1$ means that it is a member of itemset. Then,

$$\sum_{i=1}^{m-}(x_i) = k \tag{4}$$

The above is called frequent item $k$-itemset. $x_a, x_b, \ldots, x_k$ are set as the first, second, $\ldots$ , and the $k$th non-zero items from the left to right in $(x_1, x_2, \ldots, x_{m-})$, namely, $1 \leq a < b <\leq k$. According to the overall algorithm flow in Section 2.3, the Expanding Operation will be invocated for $m^-$ times (the number of nested invocation times not included), where it will be invocated in the form of Expanding Operation ($\{a\}$, $P = a$) for the $a$th time.

As shown in Figure 2, $a < b^- < b < b^+ < \ldots < k^- < k < k^+$, E-Operation ( ) represents the function of Expanding Operation. According to the principle of Expanding Operation stated in Section 2.2, the Expanding Operation is invocated by $m^-$ times. When it is invocated for the $a$th time, it is certainly judged that items $b$ and $c$ can be added into the itemset. In one step, item $b$ is added, followed by the recursive invocation of Expanding Operation ($\{a, b\}$, $P = b$). Similarly, the recursive invocation of Expanding Operation ($\{a, b, c\}$, $P = c$) is carried out after the addition of item $c$ inside the Expanding Operation ($\{a, b\}$, $P = b$). By parity of reasoning, the recursive nesting at the deepest layer is implemented by invocating the Expanding Operation ($\{a, b, c, \ldots, k\}$, $P = k$), and $\{a, b, c, \ldots, k\}$ is determined as the MFI. Hence, the MFI $\{I_1, I_2, \ldots, I_k\} = \{a, b, c, \ldots, k\}$ will be certainly found. If any MFI without loss of generality can be found, all MFIs can be too, so this algorithm ensures the high integrity of the solution set.

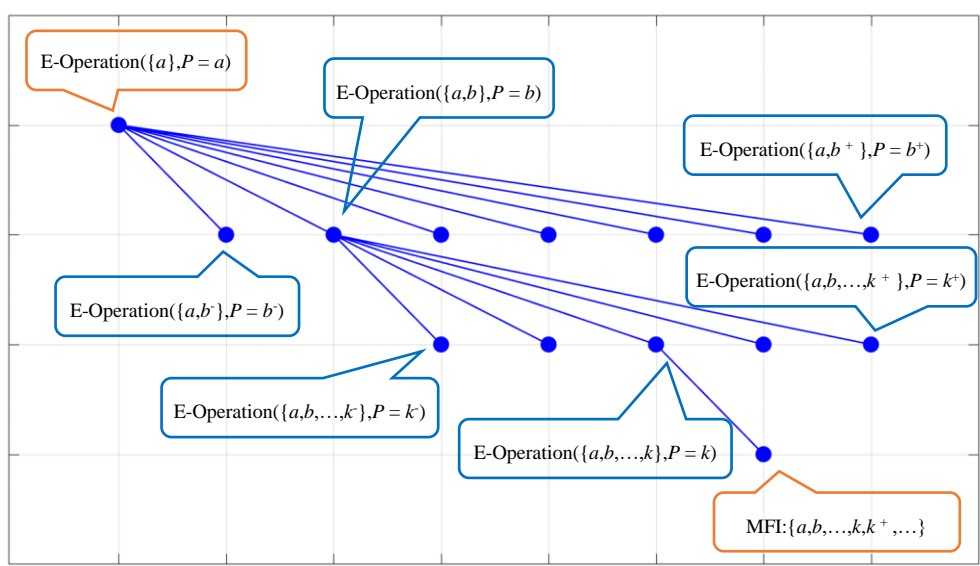

**Figure 2.** Recursive nested structures of Expanding Operation ($\{a\}$, $P = a$).

Of course, not only Expanding Operation ($\{a, b\}$, $P = b$) will be invocated inside Expanding Operation ($\{a\}$, $P = a$), but Expanding Operation ($\{a, c\}$, $P = c$) may also be parallelly invocated. Therefore, the finally found MFI after the ending of Expanding Operation ($\{a, c\}$, $P = c$) will contain $a$ and $c$ but not $b$, which is another MFI, but this is not contradictory with the hypothetical proposition to be proven.

### 3.2. Uniqueness Proof

According to the overall algorithm framework and item addition rules of Expanding Operation, each time the Expanding Operation adds one item since one frequent 1-itemset, it will be recursively invoked by itself once again or find the other items that can be added. The discovery process of an MFI is as shown in Figure 3a. The top–down sequence of nodes in the figure denotes the sequential order of item addition, the left or right position of each node represents the *ID* value of item, and the left-to-right direction is the increasing direction of *ID* value. Given this, the root nodes are located at the upper left corner and leaf nodes at the lower right corner, and this path forms a branch. The set of *ID* values of all nodes on one branch corresponds to one frequent itemset, which may be maximal or not.

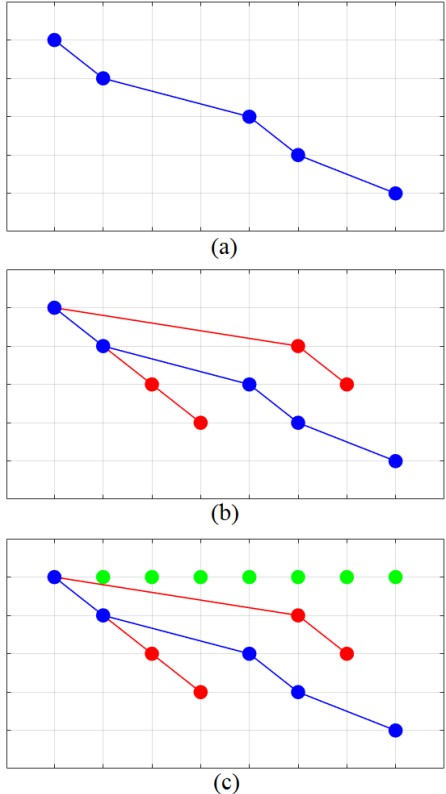

**Figure 3.** (**a**) A branch is a frequent itemset; (**b**) a tree is a group of MFIs; (**c**) the forest is the pool of MFIs.

At each root node, multiple branches may grow, so as to form a tree, as shown in Figure 3b. The *ID* values of all nodes on one branch from root nodes to leaf nodes constitute a frequent itemset, and all MFIs on one tree form a group of MFIs, as mentioned in Section 2.3.

As stated in Section 2.3, each frequent 1-itemset can serve as the root node to grow a tree. In Figure 3c, all green nodes can be root nodes to grow a tree. For the conciseness of this diagram, the tree growing out of green nodes was not drawn. Many trees form a forest, corresponding to the pool of MFIs in Section 2.3. Therefore, to prove that all MFIs are different in the pool of MFIs, it is only necessary to prove that only one branch, but not a second branch, corresponds to each itemset.

This proposition was proven through the reduction to absurdity. Assume that an MFI ($MFI_1$) is determined for a branch ($Branch_1$), and this MFI has a corresponding relation with another branch ($Branch_2$).

According to the right-hand side item addition rules of Expanding Operation, only items with greater *ID* values are added, and then the difference (difference value of x-

coordinates) of *ID* values between two nodes connected by any edge in $Branch_1$ is certainly greater than 1, while the difference value of y-coordinates is bound to be 1.

In order for $Branch_2$ and $Branch_1$ to correspond to the same itemset, the nodes of $Branch_2$ should be different from those of $Branch_1$ only in y-coordinates; otherwise, their horizontal projections will be different, so will the corresponding itemsets.

If the nodes of $Branch_2$ are different from those of $Branch_1$ only in the upper or lower position, there will certainly be such an edge in $Branch_2$ that the difference value of y-coordinates of two nodes it connects is equal to zero or greater than 1 because the difference of *ID* values between two nodes connected by the edge in $Branch_1$ is certainly greater than 1, the difference value in y-coordinates is equal to 1, and the two branches must share the same root nodes. If the difference value is equal to zero, the Expanding Operation adds two items once. If it is greater than 1, the Expanding Operation enters recursive invocation without adding any item.

Obviously, this contradicts with the rules of Expanding Operation. Then, the hypothesis does not hold true, and the proposition of path uniqueness is proven. In other words, the pool of MFIs does not contain the same itemsets. According to the proposition proven in Section 3.1 (i.e., there are no MFIs missing in the pool of MFIs), the integrity and path uniqueness indicate that the algorithm is accurate without complexity or redundancy in the path optimization.

## 4. Experiment

The algorithm proposed in this study aimed to find all MFIs, being different from most of the present algorithms aiming at frequent itemsets. The second objective was to realize standalone operations specific to big data and mass transaction sets and get rid of limitations of special computing platforms (it was thought in this study that special platforms only targeted at task decomposition to make big data computation feasible, while the computational complexity was not changed). On this basis, no horizontal comparison was made with the existing algorithms. The standalone test of only 10 open transaction sets was implemented to demonstrate the feasibility and practicability of standalone computation for big data. In the end, the detailed computation results were given, and the details of MFI with the maximum support searched by the algorithm were expounded. Readers may verify according to the results and compare with their own algorithm results. These open transaction sets are available on http://fimi.uantwerpen.be/data/ for free (accessed on 28 February 2020). The algorithm source code in this study may also be acquired from the corresponding author.

The standalone operation was implemented using the following hardware: Intel® Core™ i7-8550U CPU @ 1.80 GHz (8 CPUs) and 16,384 MB RAM, Timi Personal Computing Limited, Beijing, China. The programming language of the algorithm is Python. The advantage of using Python is that the programming environment is lightweight and easy to popularize.

### 4.1. Brief Description of Transaction Sets

The 10 transaction sets varied in volume and features. The number of transactions might reach as many as 990,000, the maximum length of transaction was 2498, and the maximum dimension was 41,270. The experimental results were given separately in two tables, where Table 2 lists the test results of small-scale transaction sets and Table 3 presents the test results of slightly larger-scale transaction sets. The results in Table 2 demonstrated the integrity of calculation results for the 10 transaction sets. Moreover, the calculation results of small-scale problem were convenient for readers to verify their algorithm accuracy. The results in Table 3 manifested the feasibility of the algorithm in the standalone operations for big data and high-dimensional transaction sets. A proper support threshold was used for each transaction set. As the data sparsity varied among the different transaction sets, they were different in the intensity of data association, and the support thresholds given in the experiment might also be greatly different.

**Table 2.** Experimental on small size transaction sets.

| 1 | Name of Transaction Set | *Chess* | *Connect* | *Mushroom* | *T10I4D100K* | *Retail* |
|---|---|---|---|---|---|---|
| 2 | Dimension of transaction set | 75 | 129 | 119 | 870 | 16,470 |
| 3 | Number of transactions | 3196 | 67,557 | 8124 | 100,000 | 88,162 |
| 4 | Average transaction length | 37.000000 | 43.000000 | 23.00000 | 10.102280 | 10.30575 |
| 5 | Minimum transaction length | 37 | 43 | 23 | 1 | 1 |
| 6 | Maximum transaction length | 37 | 43 | 23 | 29 | 76 |
| 7 | Maximum support of item | 0.999687 | 0.998757 | 1.000000 | 0.078280 | 0.574794 |
| 8 | Support threshold | 0.950000 | 0.965000 | 0.520000 | 0.022500 | 0.001250 |
| 9 | Dimension of transaction set reduced | 9 | 14 | 12 | 125 | 1652 |
| 10 | Number of transactions reduced | 3196 | 67,557 | 8124 | 97,844 | 85,840 |
| 11 | Average transaction length reduced | 8.809449 | 13.813446 | 8.886017 | 4.286119 | 7.196249 |
| 12 | Transaction set reduction rate | 0.238093 | 0.321243 | 0.386349 | 0.415125 | 0.679884 |
| 13 | Minimum transaction length reduced | 5 | 11 | 4 | 1 | 1 |
| 14 | Maximum transaction length reduced | 9 | 14 | 11 | 16 | 47 |
| 15 | Pretreatment time (s) | 2.09 | 50.33 | 5.17 | 51.33 | 46.35 |
| 16 | Algorithm running time (s) | 0.51 | 137.86 | 2.30 | 15.71 | 956.72 |
| 17 | Number of MFIs | 11 | 37 | 11 | 125 | 2586 |
| 18 | Item list of the top MFI | 29 40 52 58 60 | 75 88 91 109 124 127 | 34 36 85 86 90 | 368 | 39 48 14098 |
| 19 | Support of the top MFI | 0.969650 | 0.972231 | 0.772033 | 0.078280 | 0.006125 |

**Table 3.** Experimental on large size transaction sets.

| 1 | Name of Transaction Set | *Accidents* | *Pumsb_Star* | *Pumsb* | *T40I10D100K* | *Kosarak* |
|---|---|---|---|---|---|---|
| 2 | Dimension of transaction set | 468 | 2088 | 2113 | 942 | 41,270 |
| 3 | Number of transactions | 340,183 | 49,046 | 49,046 | 100,000 | 990,002 |
| 4 | Average transaction length | 33.807892 | 50.482139 | 74.00000 | 39.605070 | 8.099999 |
| 5 | Minimum transaction length | 18 | 49 | 74 | 4 | 1 |
| 6 | Maximum transaction length | 51 | 63 | 74 | 77 | 2498 |
| 7 | Maximum support of item | 0.999906 | 0.790054 | 0.997920 | 0.287380 | 0.607447 |
| 8 | Support threshold | 0.500000 | 0.275000 | 0.800000 | 0.025000 | 0.002400 |
| 9 | Dimension of transaction set reduced | 24 | 63 | 25 | 545 | 431 |
| 10 | Number of transactions reduced | 340,183 | 49,046 | 49,046 | 100,000 | 938,824 |
| 11 | Average transaction length reduced | 18.632792 | 31.831057 | 23.51786 | 35.122340 | 4.815123 |
| 12 | Transaction set reduction rate | 0.551137 | 0.630541 | 0.317809 | 0.886814 | 0.563729 |
| 13 | Minimum transaction length reduced | 5 | 19 | 11 | 4 | 1 |
| 14 | Maximum transaction length reduced | 24 | 42 | 25 | 67 | 398 |
| 15 | Pretreatment time (s) | 281.45 | 60.70 | 84.79739 | 123.59 | 463.31 |
| 16 | Algorithm running time (s) | 8530.40 | 245,644.41 | 46,591.14 | 645.86 | 12,645.81 |
| 17 | Number of MFIs | 216 | 512 | 3145 | 1061 | 1265 |
| 18 | Item list of the top MFI | 12 16 17 18 21 27 29 31 43 | 2297 4933 4937 7072 | 170 180 184 188 4426 4428 4430 4432 4434 4438 7062 | 54 | 1 6 11 90 148 218 |
| 19 | Support of the top MFI | 0.560634 | 0.344207 | 0.820862 | 0.088950 | 0.005552 |

## 4.2. Mining Results

Tables 2 and 3 were of the same structure. The eighth row (Support threshold) divided each table into two parts: the upper part and lower part, where the former presented the parameters of the original transaction set and the latter gave the results after the preprocessing and algorithm operation.

The name of each transaction set is given in the first row, and the dimension of each transaction set is listed in the second row, namely, the number of all different items in each transaction set. During the itemset mining process, the acceptance or rejection of these different items in the itemset constituted a solution space of the algorithm, so it was called dimension. The number of transactions in each transaction set and average length of overall transactions are presented in the third and fourthth rows, respectively, and their product could reflect the volume of each transaction set, with a direct impact on the time spent by the algorithm in traversing the CPU of the transaction set. The fifth and sixth rows list the lengths of shortest and longest transactions in each transaction set, respectively. The seventh row shows the maximum probability of single item to appear in each transaction set, which was also called the maximum support of frequent 1-itemset, reflecting the data sparsity of the transaction set.

In the eighth row, the parameters of the reduced transaction set obtained by preprocessing the original transaction set are displayed. The 9th row shows the dimension of the reduced transaction set. This dimension was decided by the given support threshold: the greater the threshold, the less the intercepted frequent 1-itemsets, the smaller the dimension, and the smaller the solution space after the reduction, which was better for

the optimization. However, fewer MFIs were found, so this was contradictory. However, Table 3 shows that the dimensions of transaction sets reduced from the two largest transaction sets were taken as 545 and 431, respectively, and the algorithm running time was also acceptable. The number of transactions in each reduced transaction set and the average length of overall transactions are given in the 10th and 11th rows. If the two figures were reduced, the volume of each transaction set was also shrunk. The shrinkage rate (12th row) was acquired by dividing two figures, where the divisor was the product between the number of transactions in the original transaction set and the average transaction length, and the dividend was the product between the number of transactions in the reduced transaction set and the average transaction length. Although the shrinkage degree of the transaction set was also decided by the given support threshold, individual transactions in each transaction set had different properties, so the shrinkage rate would differ even under the same threshold. The minimum and maximum transaction lengths of reduced transaction sets are given in the 13th and 14th rows.

The preprocessing time of transaction (15th row) was the time spent on acquiring a reduced transaction set from the original one, including the time needed to acquire the parameters of the original transaction set, generate files of the reduced transaction set, and establish a memory data matrix. The time(s) needed to mine all MFIs is presented in the 16th row. Table 3 shows that the number of transactions in the largest transaction set *kosarak* reached nearly million class, where its original dimension was 41,270. When the dimension was reduced to 431, it took the algorithm less than 4 h to find all MFIs. The quantity of MFIs found was 1265, which, theoretically, was accurate (i.e., the sets were of integrity and uniqueness). Therefore, the standalone big data association rule mining was feasible. If the transaction sets came from practical production and life, the algorithm would be pragmatic.

The 17th–19th rows list the number of MFIs found, item details in the MFI with the maximum support, and the corresponding support degree, respectively, aiming to facilitate readers in verifying and comparing this information.

### 4.3. Comparison of Algorithm Running Time with Solution Space Size

From the above experiment, a reduced transaction set was acquired from each original transaction set after the support threshold was given, and the dimension was also reduced. Therefore, the reduced dimension would vary with the support threshold, and so would the algorithm running time. The algorithm was proposed to implement the standalone operation of big data transaction sets. For the same transaction set, the reduced dimension would be increased if the support threshold was reduced, and as a result, the solution space would present an exponential (base number: 2) increase. If the algorithm running time was also increased according to an exponent with a base number of 2, the algorithm running time would experience explosive growth with the increase in dimension, which, obviously, deviated from the intention of this study.

To this end, the largest transaction set *kosarak* was experimented under different thresholds for five times. The dimension reduced grew from 42, reaching as high as 431, and the acquired number of MFIs and algorithm running time are as seen in Table 4. The data showed that the algorithm running time did not present exponential growth with the increase in dimension.

**Table 4.** Testing on *kosarak* with a different support threshold.

| Support Threshold | 0.0127 | 0.0063 | 0.0040 | 0.0029 | 0.0024 |
|---|---|---|---|---|---|
| Dimension reduced | 42 | 106 | 212 | 328 | 431 |
| Number of MFIs obtained | 65 | 212 | 467 | 876 | 1265 |
| Algorithm running time (s) | 238.9 | 1054.2 | 2962.1 | 6714.2 | 12,645.8 |

The solution space size was incomparable to the algorithm running time in dimension. As the algorithm ran for 238.9 s under the dimension of 42, the dimension-dependent

exponential growth curve of solution space took the point (42, 238.9) in the plane as the starting point, and the fitted curvilinear contrast relation between the growth curve of solution space and actual algorithm running time is as shown in Figure 4. Obviously, when the solution space showed exponential explosive growth (red line in the figure) with the dimension, the algorithm running time was under a slow growth trend (blue line with red circle, the experimental data are shown at the red circles). As the algorithm running time did not present exponential growth with the dimension (i.e., it did not experience explosive growth), the algorithm was feasible for the standalone calculation of high-dimensional properties of big data transaction sets.

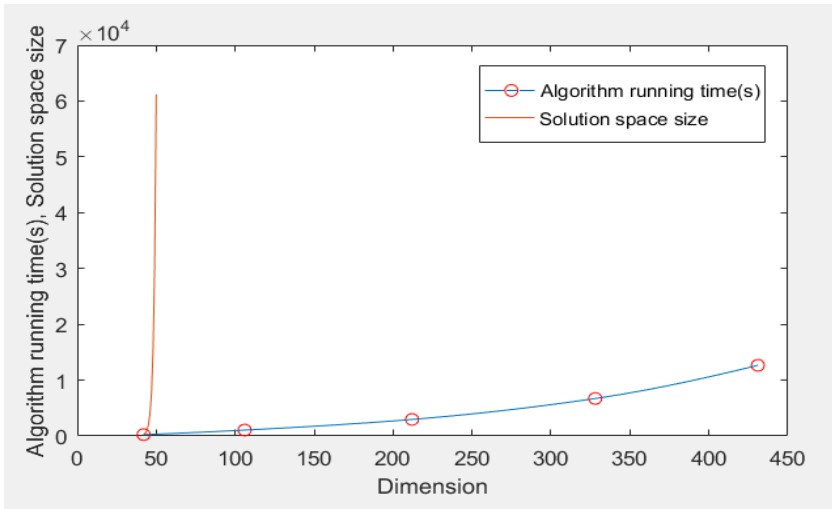

**Figure 4.** Solution space size compared with the algorithm running time.

### 4.4. Comparison of Algorithm Running Time with Traditional Exact Algorithm

The Apriori and FP-Growth algorithms were the two most classical accurate algorithms for mining association rules. PF-Growth was recognized to be faster than Apriori. However, PF-Growth was sensitive to the dimensions of the transaction set. Even the running time is unacceptable for mining the small transactions set listed in Table 2, but the smallest dimension of the transaction set listed in Table 2 is only 75.

In order to investigate quantitatively the running time of the algorithm, we extracted a small part of the Accident transaction set on average, such as 100, or 200, or 300 transactions, etc. Each transaction captures only 10%, or 20%, or 30% at the beginning of each transaction. In this way, small transaction sets with different transaction numbers and dimensions are used to test the running time of the PF-Growth algorithm. The dimensions of each transaction set are listed in Table 5. The rows represent the different number of transactions, and the columns represent the percentage of each transaction captured. These transaction sets were used to test the PF-Growth algorithm, and the running time of PF-Growth is shown in Table 6.

**Table 5.** Dimensions of extracted *Accidents*.

|      | 100 | 200 | 300 | 400 | 500 |
| --- | --- | --- | --- | --- | --- |
| 10% | 15  | 16  | 17  | 17  | 17  |
| 20% | 22  | 24  | 25  | 26  | 24  |
| 30% | 29  | 31  | 30  | 33  | 31  |

**Table 6.** Running time of FP-Growth for the extracted *Accidents*.

|       | 100            | 200            | 300            | 400            | 500            |
|-------|----------------|----------------|----------------|----------------|----------------|
| 10%   | 13.116257 s    | 18.344052 s    | 26.346489 s    | 40.406259 s    | 44.279161 s    |
| 20%   | 444.653654 s   | 714.085102 s   | 772.977960 s   | 901.212257 s   | 982.893483 s   |
| 30%   | 7956.674653 s  | >2.21 h        | >2.21 h        | >2.21 h        | >2.21 h        |

Since FP-Growth algorithm mined frequent itemsets and proposed algorithm mined maximal frequent itemsets, it had been verified that the results of FP-Growth were indeed subsets of the maximal frequent itemsets mined by proposed algorithm.

Table 6 reflects that the FP-Growth algorithm was not sensitive to transaction number but sensitive to dimension. When the dimension grown from 15 to 29, the FP-Growth had a longer running time than the exponential growth. By comparison, the proposed algorithm mined the full size set of incident transactions in only 8530.40 s. Of course, the proposed algorithm had fewer optimization targets, while FP-Growth had more optimization targets. The former was the maximal frequent itemsets, while the latter is the frequent itemsets, and the number of maximal frequent itemsets was less than the number of frequent itemsets.

*4.5. Comparison with FPMax, CD, and DMM*

Since parallel algorithms were very promising in solving big data mining, we make a simple comparison between the proposed algorithm and several well-known parallel algorithms, which are FPMax algorithm [38], Count Distribution algorithm [37], and DMM algorithm [40]. Although PMM algorithm [39] was also an excellent parallel algorithm and DMM was better than PMM algorithm overall, we do not compare with PMM.

In Section 4.2, we present some experimental results on 10 public datasets. Some of these datasets have also been reported in the literature [37,38,40]. We picked the experimental results data to the same dataset as Table 7. Some figures were estimated from reported graphs, which gave a rough idea of algorithm performance.

**Table 7.** Running time and resources used for various algorithm on various databases.

|                    | *Chess* (0.95) | *Mushroom* (0.52) | *Pumsb* (*)    | *Accidents* (*) |
|--------------------|----------------|-------------------|----------------|-----------------|
| FPMax              | <0.4 s         | 0.3 s             |                |                 |
| CD                 |                |                   | 35,000 s       | 28,000 s        |
|                    |                |                   | 0.8, node:8    | 0.4, node:1     |
| DMM                |                |                   | 9000 s         | 3000 s          |
|                    |                |                   | 0.8. node:8    | 0.4, node:1     |
| Proposed algorithm | 0.51 s         | 2.3 s             | 46,591.14 s    | 8530.40 s       |
|                    |                |                   | 0.8, node:1    | 0.5, node:1     |

With regard to the *Chess* dataset and *Mushroom* dataset, the experimental results of FPMax and our algorithm had the same support thresholds of 0.95 and 0.52. The running time of the algorithm is listed in Table 7. FPMax is faster than the proposed algorithm, but the former was conducted using a cluster of computers, while the latter was carried out by a single computer. Of course, the hardware of the computer is also different, and the small data set leaded the running time to be haphazard.

The *Pumsb* dataset with the same support threshold 0.8 to three algorithms, CPU-time of CD algorithm was approximately 35,000 s, our algorithm was 46591.14 s. CD is great in that it executed the *Pumsb* * dataset, which is eight times larger than the *Pumsb* dataset. However, CD was executed on an eight-node cluster calculation. The DMM execution time under the same conditions was 9000 s, which was faster than CD algorithm. Additionally, DMM was executed on an 8-node cluster calculation.

The *Accidents \** dataset is four times larger than *Accidents*. To *Accidents* dataset and *Accidents \** with the different support threshold, CD and DMM for *Accidents \** with 0.4 and our algorithm for *Accidents* with 0.5. All three algorithms use single machine, that is, the number of nodes is 1. DMM was the fastest, CD was the slowest and our algorithm was in middle. However, the support thresholds they used were different. If the threshold value was small, more MFIs would be mined and it would take a longer time.

To sum up, the running time is basically an order of magnitude, and each algorithm has its own advantages and disadvantages. Some are suitable for dense datasets, while others are for long transaction data sets. Under the limitation of computer hardware, it is feasible to select the proposed algorithm which was executed on single notebook computer.

## 5. Discussion, Conclusion, and Expectation

According to the algorithm theory stated in this study, the transaction sets should be visited during the MFI searching. Therefore, the larger the volume of transaction sets, the longer the time needed by the algorithm running. The volume of the transaction set is the product between the quantity of transactions and average transaction length. As the Expanding Operation needs to add items, the greater the dimension of the transaction set, the longer the time needed by the algorithm running.

However, in the experiment on the 10 transaction sets, the reduced dimensions of both *pumsb_star* and *pumsb* are no more than 63, which are small compared with those of other transaction sets in the table, but the two transaction sets take the longest time (at least 12 h). The reduced dimension of *T40I10D100K* is the maximum (545), but it takes the least time, about 10 min. Moreover, the quantity of transactions and average transaction length in *T40I10D100K* are greater than those in *pumsb_star* and *pumsb*, and the quantity of transactions is even more than twice of those in *pumsb_star* and *pumsb*.

This is just like picking fruits in an orchard: the larger the orchard, the longer the time needed to search it once, but the experimental data show that this is not the case. Then, is it the case that the more the fruits, the longer the time needed? However, the experimental data in Table 3 show that the quantity of MFIs found in *T40I10D100K* is larger than those found in *pumsb_star* and *pumsb*. Is this related to the length of MFIs found? Even if the average length of MFIs found in *T40I10D100K* is multiplied by 10 in order that it is equivalent to the average length of MFIs found in *pumsb_star* and *pumsb*, the calculation time will also be tenfold. Even so, the time spent is less than 1/10 of the time spent by *pumsb_star* and *pumsb*.

On the basis of the above analysis, there should be only one possibility: the main factor influencing the algorithm running time is neither the volume of transaction set nor the quantity of MFIs found, but it is correlated with the complexity of MFIs found. The complexity of MFIs does not have a linear relation with their length. Instead, the two present a growth relation greater than the linear growth relation. Whether this relation is an exponential relation and what the exponential relation (base number) remains to be further explored.

From another side, this indicates that the algorithm running time is not sensitive to the volume of the transaction set, so the algorithm is practical for big data.

Given this, can we understand it that the algorithm spends the time mainly in "picking fruits" but not "finding fruits"? From this angle, does it mean that the algorithm spends most of the time in acquiring the optimal solution with the excellent optimization path? The long algorithm running time is ascribed to the large quantity of MFIs but not to the large volume of transaction set or the great dimension of the reduced transaction set. The practical experimental data also verify that the algorithm is feasible, and even superior, for the standalone operation of high-dimensional big data transaction sets.

The algorithm is feasible and accurate for the MFI mining of high-dimensional transaction sets. However, it has a disadvantage in the association rule analysis based on frequent itemset mining. It can be known from the dimension of the reduced transaction set under the given support threshold that the time needed to search all MFIs subsequently is un-

predictable. At times, the acceptable time is limited, and it is unnecessary to find all MFIs. Instead, the MFI with the maximum support should be first found within the limited time, followed by the MFI with the second largest support, and so on. In this way, the algorithm practicability will be higher.

Therefore, the subsequent study of this algorithm should focus on MFI mining under an adaptive support threshold in the following way: no support threshold is given in the algorithm running. Instead, as the time passes by, the algorithm finds the thresholds of all MFIs and sort them in a descending order. Within given time, the algorithm returns the minimum support threshold and the corresponding MFI.

**Author Contributions:** Conceptualization, Y.Z. and W.Y.; methodology, Y.Z., W.Y., X.M. and H.O.; software, Y.Z.; validation, Q.Z., W.Y. and Y.Z.; formal analysis, Q.Z.; investigation, W.Y.; resources, W.Y.; data curation, Y.Z.; writing—original draft preparation, Y.Z.; writing—review and editing, Y.Z.; visualization, Y.Z.; supervision, W.Y.; project administration, Y.Z.; funding acquisition. All authors have read and agreed to the published version of the manuscript.

**Funding:** This work was supported by the Zhejiang Basic Public Welfare Research Plan Projects (Item No. LGG19F030009), funded by the Science Technology Department of Zhejiang Province, China.

**Institutional Review Board Statement:** Not applicable.

**Informed Consent Statement:** Not applicable.

**Data Availability Statement:** Publicly available datasets were analyzed in this study. This data can be found here: http://fimi.uantwerpen.be/data (accessed on 21 September 2021).

**Acknowledgments:** We would also like to extend our gratitude to the website http://fimi.uantwerpen.be/data/ (accessed on 21 September 2021) for providing the open test transaction set to a wide range of researchers.

**Conflicts of Interest:** The authors declare no conflict of interest.

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
