# Peer review of "Right-Hand Side Expanding Algorithm for Maximal Frequent Itemset Mining"

_applsci, doi:10.3390/app112110399_

Round 1
Reviewer 1 Report
The author is suggested to consider a seperate literature review section. where research gaps or the research questions should be demonstrated. This research questions should be cross-checked with methodology, results and conclusion,
Author Response
Thank you for your constructive comments. We have revised the paper according to your comments one by one. Your comments have contributed greatly to the improvement of the quality of the paper. But there are limits to what we can do, and mistakes are inevitable. Please feel free to let me know if there is anything problem in paper.

Reviewer 2 Report
This paper proposed an algorithm for mining maximal frequent itemsets (MFIs). The proposed algorithm uses a strategy called Expanding Operation to expand the size of a given frequent itemset. Those supersets formed by this strategy are MFIs. The experiment was conducted on ten transaction datasets to verify the effectiveness of the proposed algorithm in terms of patterns found and computational cost.
In general, the novelty of the paper is not so high since previous works have considered the problem of mining MFIs, both sequential and parallel approaches. However, the paper is well organized and fairly presented. Moreover, since MFIM is one of the major topics in pattern mining, I still encourage the acceptance of this paper if the authors carefully revise the paper to further improve its quality.
Some parts of the paper are hard to follow even though I have worked with the topic of MFIM before. I have a serious concern that the motivation and relevance of the paper are somewhat questionable and unclear. In addition, the formulation of the proposed method is not well developed and partly violates the standard convention of mathematical writing.
- Discuss the applications of MFIM
- Discuss the motivation of the proposed algorithm, if only for the large-scale dataset, we already have several algorithms (see 10.1109/TAI.2003.1250181 or https://doi.org/10.1007/s10115-007-0115-1)
- Highlight the contributions of the paper in the Introduction
- The FPMax algorithm should be discussed in the Introduction. It is a very efficient algorithm for maximal itemset mining.
- Insert a table of notations
- Figures 1 and 2 should be revised, the current forms are not in a good shape, make the text more clear, copiable, and be not broken when zooming.
- Use a running dataset and give examples for each definition and when you explain the workflow ( or pseudo-code) of the algorithm.
- I suggest illustrating the Expanding Operation as a tree structure since the current form is hard to follow.
- [Important] Carefully check and revise notations, equations, definitions used in the paper.
- [Important] Rewrite the pseudo-code in Algorithms 1 and 2, typos can be found in both Algorithms.
- Several notations were not explained clearly, for example, what is “k” in Eq.3. Check all notations in the paper.
- In the experiment, why authors did not compare the performance of the proposed algorithm with previous works such as FPMax or 10.1109/TAI.2003.1250181. I suggest authors to make a fair comparison with those algorithms in terms of computational complexity to verify the efficiency of your proposed algorithm.
Author Response

(The authors gave the same response as above.)

Round 2
Reviewer 2 Report
This version has addressed most of my comments on the previous version. The current paper has been improved in terms of clarity and consistency. In this version, the authors should consider the following points to further improve the quality of the paper before I vote for an acceptance.
- First, as mentioned before in the previous version, authors should discuss the applications of MFIM. I mean that how MFIM can be applied to other topics in the literature or real-world applications. To this end, I suggest authors briefly introduce the applications of MFIM in outliers detection [https://doi.org/10.1007/978-3-540-78849-2_13], web clustering [https://doi.org/10.1007/978-3-642-03079-6_15] and categorical clustering [https://doi.org/10.1007/s10489-020-01677-5].
- The pseudo-code of Algorithms 1 and 2 are still not in a good shape, for instance, in the input and output, you do not need to put two lines of dots between x_1,x_2, and x_m; just like this (x_1,x_2,...,x_m).
Also, insert a section in the paper to theoretically discuss the complexity of the proposed algorithms. For your reference, you can explain the complexity by referring to [https://doi.org/10.1007/s10489-018-1227-x] in the discussion, in which they do the same analysis in pattern mining.
Author Response
Thank you for your constructive comments. We have revised the paper according to your comments one by one. Your comments have contributed greatly to the improvement of the quality of the paper. Please feel free to let me know if there is anything problem in paper.

This manuscript is a resubmission of an earlier submission. The following is a list of the peer review reports and author responses from that submission.
Round 1
Reviewer 1 Report
The authors presented an algorithm for maximal frequent itemsets mining. The problem is technically sound, and the proposed algorithm seems effective especially while working with large transaction sets. The authors justified the requirement of the proposed work in the introduction with a brief literature survey. The algorithm is well explained and the presented experimental results are satisfactory. Please see below some minor suggestions:
I advise the authors to include some references for the claims made in the first paragraph of the introduction. You are using the term “Big data association rule mining” at the beginning of the second paragraph but explaining the general concepts of association rule mining. The actual challenge of big data is introduced in paragraph 7. I suggest the authors carefully review the logical flow of ideas in the introduction.
In Section 4, the authors provided the hardware configuration that was used to generate the results. Which programming language or platform was used to implement the algorithm? It will be very helpful if such implementation is made available as a package or library in data analysis software platforms such as R or python.
The rows of the Table (1 or 2) are being referred to as 2nd row, 8th row, etc. It will be helpful to provide an extra colum in the table with row number so that readers can easily find the appropriated row in the table (otherwise you must count every time find a referred row).
The first letter of each word in the heading and subheading should be capitalized.
Author Response
Thank you so much for your constructive comments. We have revised the paper according to your comments one by one. Your comments have contributed greatly to the improvement of the quality of the paper. Mistakes are unavoidable, please feel free to let us know if any problem.

Reviewer 2 Report
The authors tried to propose an algorithm to solve the problem of time cost and platform cost used in data mining by association rules. They believed that the proposed method can correctly find the Maximum Frequent Itemsets (MFIs). The authors claim that the special design of Expanding Operation can reduce computing complexity, so it is suitable for calculations with large amounts of data. In the Final, an experimental report on 10 open standard transaction sets was given in this study, including the big data calculation results of million-class transactions. But I still have to point out the following issues related to this article.
1. The sentences in this article are not fluent in meaning, and it is easy to confuse what the authors wants to express. For example, the sentence, “This algorithm not only used any special computing platform or required cluster operation, but it could also find the exact solution—all MFIs of big data transaction set—to big data problem within an acceptable time range.” Seems make reader confused in section 1.
2. There is a lack of explanation in the content and symbols of the equation. The symbol of the algorithm is also unclear. In Section 2.1, the equation (3) should be explained in detail. ??k is not the natural numbers, since I= {I1, I2, …, Im} is the set of all items. In Algorithm 1, the pseudocode of transaction set preprocessing is expressed. But its symbol such as “for j = 1 : jTij do” which is no any description.
3. Finally, in the part of the experiment, the authors should compare with various algorithms of traditional association rules to strengthen the purpose and contribution of this article. For example, there should be some comparisons in terms of the accuracy of the final result and the time efficiency.
Author Response

(The authors gave the same response as above.)
